# The Effect of Environmental Contexts on Motor Proficiency and Social Maturity of Children: An Ecological Perspective

**DOI:** 10.3390/children8020157

**Published:** 2021-02-19

**Authors:** Zahra Fathirezaie, Kosar Abbaspour, Georgian Badicu, Seyed Hojjat Zamani Sani, Hadi Nobari

**Affiliations:** 1Physical Education and Sport Science Faculty, University of Tabriz, 29 Bahman Blvd, Tabriz 51666-16471, Iran; kosar.abbaspour@gmail.com (K.A.); hojjatzamani8@gmail.com (S.H.Z.S.); 2Department of Physical Education and Special Motricity, Faculty of Physical Education and Mountain Sports, Transilvania University of Braşov, 500068 Braşov, Romania; 3HEME Research Group, Faculty of Sport Sciences, University of Extremadura, 10003 Cáceres, Spain; hadi.nobari1@gmail.com

**Keywords:** motor skills, social quotient, ecological perspective

## Abstract

Physical environmental factors affect the developmental process of children. Thus, the main purpose of the present study was to investigate the two intervention models of affordances on the motor proficiency and social maturity of children. A semi-experimental research design with a pretest–posttest design and two groups were used, adopting the convenience method. Two groups of 15 children (aged 5.5–6.5 years) engaged in 12 weeks of nature school or kindergarten. The Bruininks–Oseretsky test of motor proficiency and the Vineland social maturity scale were used. The results of a mixed ANOVA showed that natural outdoor activity has a greater positive effect on motor proficiency and social maturity than kindergarten activities. Intra-group analysis also showed that both groups had progressed, but the nature school group made more progress. These results were discussed and interpreted based on the types of environmental affordances, Gibson’s theory, Bronfenbrenner theory, and child-friendly environment. It was suggested that natural environmental stimulations play a critical role in optimal child motor and social development during the early stages of life.

## 1. Introduction

The first years of life play an important role in the growth and development domains [1]. Thus, researchers are trying to identify the biological [2] and environmental factors that can enrich this period of life [3]. In this regard, motor development refers to the motor behavior changes influenced by biological and environmental factors, and related research focuses on changes in these contexts, which manifest as observations of evolution over time [4]. Although some biological factors are uncontrollable, environmental factors are within our control and play a major role in shaping who we become [3]. Research has demonstrated that the perceptions of children may be affected by physical environmental factors [3,5,6,7,8,9], cognition [6,10,11], and the type and rate of children’s physical activity [9,12,13,14,15,16]. In actual fact, the environment is very critical in the development of children and it represents the physical and psychological stimulation the child receives [17]. Hence, child multilateral developmental manifestations and patterns can be studied under the influence of ecological factors [3,12,18,19,20].

Hence, children’s developmental patterns are influenced by responsiveness to relations, experiences, stimuli, and settings throughout their lives, and provide situations for buffering [3]; thus, it is claimed that complex and collective bioecological contexts shape human development by the formation of societal and environmental constructions [21]. In this regard, ecological psychology theory cites that the person and the environment are regarded as mutually linked in an “eco niche” [22], such that in Gibson’s theory, the functionally significant properties of the environment are known as affordances [9,22]. Gibson’s ecological perspective and Bronfenbrenner’s ecological systems theory have close relevance to the study of motor development. These two theoretical models consider motor development to be the result of related processes between the child and his or her proximate settings, which can be assessed by the available affordances for growing movements [23]. Affordances are defined as environmental goods for certain capabilities of a person [8,9,24]. Based on this conceptualization, the notion of affordance is related to a child’s perception of environmental properties [25]. Because preschool is so important for a child’s development, what the environment of preschools provides and what is perceived or recognized by children as realizable in relation to their needs, interests, motivation, or capabilities are known as affordances [23,25]. Moreover, affordances are the functional significances of environmental characteristics for an individual and the characteristics of the environment that are real and substantial, both objectively and psychologically [22], as well as the properties of the environment that are functionally significant. They are perceived by way of active information detection and they include dimensions of the environment and the individual [26]. Therefore, based on the encouraging ability of affordances in a given space, play may differ from one child to the next [22,25,26].

Consequently, the significance of building outdoor environments, in which a broad diversity of affordances is available and in which the range of properties affords play and detection, could enable children to participate in various play behaviors [25] and interactions that increase their developmental potential [27]. Some researchers have shown the effect of physical and social affordances on children’s physical activity [28,29], social play [25], and risky, exciting, and intense play opportunities [9]. According to the theory of affordances [30], the properties of the physical environment can be hypothesized to influence children’s play behaviors [8]. Moreover, environmental factors such as physical, social, or symbolic may request, allow, or prohibit bilateral settings, as for a close complex activity. These interlay procedures of progress are dependent on the interrelation between the person and the environment. Indeed, Gibson’s theory of affordances also emphasizes the same person–environment relationship [23]. Thus, if the purpose is to increase children’s development, then promoting children’s play through a supportive environment might be a beneficial method, and the preparation of interventions to promote the physical environments experienced by children across the globe is warranted [17].

Since weighty circumstances with a diversity of choices and possibilities for allowing children to follow their interests supply the best environments for learning [7], a developmental and pedagogical context and outdoor environments, across the extra inspection of environmental properties with a functional emphasis on the child, could facilitate children’s play, learning, and development [23,31].

In this regard, natural surroundings allow children to probe their environments, experience new things, engage in challenges, and take risks [25]. One of the spaces that provides this opportunity for children is nature schools or forest schools/kindergartens, an approach constructed by categorizing children’s play into a more specific and different play that is known for its unstructured and nature-based play [32]. Playing in such spaces is accompanied by an increase in the prevalence of physical movement, and growth of cognitive development and critical social skills had been reported [10,33]. While typical preschools provide structured classes for children, due to the different methods in nature schools, the main aim of this study was to investigate the effect of different environmental contexts, such as indoor (kindergartens) and outdoor (nature schools) spaces, on the motor proficiency and social maturity of preschool children.

## 2. Materials and Methods

### 2.1. Subjects and Design

This research was quasi-experimental in nature and included two groups (15 children aged 5.5–6.5 years in each group) with a pretest–posttest design, conducted with purposive sampling in Tabriz, Iran. The first group included nine girls and six boys, named the outdoor physical activity group, who participated in the Nature School of the University of Tabriz. The second group included eight girls and seven boys, who were registered in the normal kindergarten of the University of Tabriz and participated in the present study as the indoor physical activity group, meaning that they did not experience playing in the natural environment outside. At first, a letter of satisfaction was received from parents for their children’s participation in the research, and all participants voluntarily took part in the present study. Additionally, no behavioral, motor, or cognitive problems were reported for any of the children.

### 2.2. Procedure

The intervention was conducted in 12 sessions and each session lasted approximately 4 h. The intervention of the outdoor physical activity group included nature play. In this space, children were free to explore different ways of feeling, behaving, and interacting with others. They could have more space, carefree about their clothes getting dirty and speaking out loud. There were also special spaces in the nature school, which included natural space, adventure space, active play space, quiet play space, and solitude space, which were created to promote and encourage play in nature. These spaces had natural elements such as sand, soil, water, wood, living beings, stone, fire, etc. and open and shielded spaces, different parts that can be wielded by children, and the possibility of ‘chance’ events. The children’s activity in the nature school could not be controlled from session to session due to the lack of a definite schedule and the optionality of the activities, but the total attendance hours were considered the same in both conditions (16 h per week).

The second group of the study consisted of children enrolling in the typical kindergarten, those who performed their activities in the indoor space of the kindergarten and for each activity there was a specific timetable, 4 days per week, including painting (3 sessions per week), math learning (2 sessions per week), language learning (3 sessions per week), poetry reading (3 sessions per week), and play by presenting a predetermined pattern (3 sessions per week) and playing dough (2 sessions per week).

Additionally, the children in this kindergarten did not have any outdoor experiences. The main activities of the children in the kindergarten included painting training, curricula, and crafts.

### 2.3. Instruments

The Bruininks–Oseretsky test of motor proficiency (BOTMP) was used to assess the children’s gross and fine motor skills. The BOTMP consists of eight subtests (running speed and agility, balance, bilateral coordination, strength, upper-limb coordination, response speed, visual–motor control, and upper-limb speed and dexterity), with a total of 46 separate items that provide a comprehensive index of motor proficiency, as well as separate measures of both gross and fine motor skills. Four of these subtests combine to form the gross motor composite score, and three subtests combine to form the fine motor composite score. These seven subtests, together with a subtest of upper-limb coordination, form the battery composite score. The BOTMP provides age-norm, composite scores, based on a T-score with a mean of 50 (SD = 10). This test has wide clinical and educational acceptance because it measures the skills important to children’s development, and it is perceived to have good psychometric properties. Moreover, the validity and reliability coefficients of this scale have been approved [34,35].

The Vineland Social Maturity Scale (VSMS) was originally developed by Doll in 1935, which was then adapted by Malin in the year 1965 to measure social maturation in eight social areas. The VSMS is one of the main psychological assessment tools for the evaluation of social and adaptive functions. It measures the differential social capacity of an individual, including self-help genera (15 questions), self-help eating (12 questions), self-help dressing (13 questions), self-direction (14 questions), occupation (22 questions), locomotion (10 questions), communication (14 questions), and sociality (17 questions). Moreover, it provides an estimate of social quotient. The items are arranged in order of “normal average life age progression” and in increasing order of difficulty. It is meant to measure maturation in social independence or social competence from infancy to young adulthood, from age 0 to 25. The main purpose of each item is to represent a particular aspect of the ability to look after one’s own practical needs. Doll used a “direct total score conversion” table to arrive at the social age [36].

### 2.4. Data Analysis

Using IBM Corp. Released 2015. IBM SPSS Statistics for Windows, Version 23.0. Armonk, NY: IBM Corp and Microsoft Excel (2013), the mean, standard deviation (SD), and figures were explored and illustrated. Missing data, normality of the data distribution, and group differences at the pre-test stage were checked. The data were evaluated in terms of normal distribution using the Shapiro–Wilk test. A mixed ANOVA analysis of variance was used to examine the research hypotheses. For all analyses, the level of significance was set at alpha < 0.05.

## 3. Results

Descriptive statistics of motor proficiency and its subscales, gross and fine motor skill scores, social function score, and social quotient are illustrated in Figure 1, Figure 2 and Figure 3.

The multivariate test of the mixed ANOVA showed that time (pretest-posttest) main effect (Wilks’ Lambda value = 0.26, F = 80.69, *p* = 0.0001, partial eta squared = 0.74), motor main effect (Wilks’ Lambda value = 0.02, F = 106.55, *p* = 0.0001, partial eta squared = 0.98), and the interactive effects of time *groups *motor skills were significant (F = 21.97, *p* < 0.0001, partial eta squared = 0.920).

Meanwhile, t Mauchly’s test showed that the spherical assumption did not meet (*p* ≤ 0.05). Thus, we reported the Greenhouse–Geisser parameter for the group main effect (type III sum of squares = 4709.34, mean square = 4709.347, F = 373, *p* = 0.002, partial eta-squared = 0.306) and the train*motor*group interaction effect (type III sum of squares = 6434.57, mean square = 2749.74, F = 22.59, *p* = 0.0001, partial eta-squared = 0.44). The results of the post-hoc test are shown in Table 1 and Table 2.

The multivariate tests showed that the interactive effects of the test stages, intervention groups, and social development were significant (F = 16.01, *p* < 0.0001, partial eta-squared = 0.859).

Further analyses using Mauchly’s test showed that the spherical assumption was not met (*p* ≤ 0.05). Thus, we reported the Greenhouse–Geisser parameter for the group main effect (type III sum of squares = 1487.39, mean square = 1487.39, F = 34.72, *p* < 0.0001, partial eta-squared = 0.554) and the train*motor*group interaction effect (type III sum of squares = 5637.78, mean square = 5382.19, F = 7.87, *p* = 0.009, partial eta-squared = 0.219). The results of the post-hoc test are shown in Table 3 and Table 4.

## 4. Discussion

We conducted this study to examine the effect of environmental contexts (i.e., affordances) on the motor proficiency and social maturity of children. Our findings highlighted that natural outdoor activity has positive and significant effects on motor proficiency, running speed and agility, balance, bilateral coordination, strength, gross motor skills, upper-limb coordination, response speed, visual–motor control, and fine motor skills. The results also showed that the activity of children in indoor spaces has a positive and significant effect on their visual–motor control skills. Moreover, the differences between the two groups showed that children who were active in the natural open environment made more progress in terms of running speed and agility, balance, bilateral coordination, strength, gross motor skills, upper-limb coordination, visual–motor control, upper-limb speed and dexterity, fine motor skills, and motor skills than the children who experienced the indoor environment. Our findings in social development showed that natural outdoor activity has a positive and significant effect on all social developmental factors, while indoor space activity (kindergarten) had a significant effect on social quotient, communication, and self-help genera. Moreover, the differences between the two groups showed that the children who were active in the natural environment made more improvement in their social quotient, self-help genera, self-direction, occupation, locomotion, and sociality than the indoor environment activity.

Other research has noticed the effect of outdoor environments on children’s development [4,25,31,37,38]. A closer look at pedagogy shows that different authors have also cited that the nature of these environments and activities that make children more active are very important components [13,37]. In this regard, ecological psychology states that children actively learn as part of an interconnected system by exploring their environment. For children, perceptual experiences, awareness of the structure of objects, and discovery of objects are important. It seems that natural environmental affordances prepare functional facilities and have psychological concepts [39]. We are in a sociocultural environment where there is a certain normative approach to the use of objects, in that such devices are designed for a specific purpose and seriously limit our behavior. For example, chairs are for sitting, although we may use them in many other ways. Additionally, the equipment available at clubs or other venues where people play sports provides such standardized equipment [40]. On the other side, in natural environments, such canonical affordances seem relatively absent. For instance, there is no specifically prescribed way in which we ought to use a tree trunk that we encounter in a forest. We can climb on it, kick it, jump from it, walk over it, push it, hug it, etc. Hence, natural environments seem to allow and invite more diverse behaviors than manufactured ones [41]. Additionally, a rock offers chances to hide and climb, but not all people who encounter it will choose to climb on it or hide behind it. Some others may climb on the rock to pretend a bird, so they experience various psychological states. A variety of children’s experiences occur, while adults have a more rational view of a rock, they usually use it to experience relaxation or physical activity. Thus, by defining the environment in terms of the person, the transactional approach provides a meaningful description of the environment, capable of capturing the characteristic relational properties of human experiences [39]. Therefore, affordances could be extracted from the different play activities from one child to the next in a given space. Therefore, some authors have emphasized the importance of designing environments in which a vast diversity of affordances are available and in which play and exploration are supplied and implemented. As children can gain opportunities for their development and learning by performing different play behaviors and interactions, this style fits perfectly with “child-friendly environments.” It is a condition that prepares more chances and challenges for children’s play and interactions [7,25]. It seems that nature schools could be introduced as a function of these concepts. In this research, the nature school represented a place in which learning took place in natural spaces and in sensory, experiential, and kinesthetic forms. That is, a school where achievement, curiosity, development, and inspiration transpire in environments that are not cultivated by humans. In such environments, children learn from the environment around them, thereby improving their key developmental skills, including (but not limited to): cognitive, social–emotional, physical–motor, observational, and creative thinking [32]. Within this nature school, students were given the opportunity to learn through hands-on activities, exploring and experiencing the nature around them, and taking risks [3,31]. It has been shown that educational programs combining outdoor activities in early childhood with direct, ongoing experiences of nature in relatively intimate circumstances can be introduced as essential principles for the comprehensive and multifaceted development of children [42].

According to Gibson [30], affordances are opportunities for an individual’s functioning that are only understood by the person to make a person–environment fit. Thus, places, objects, events, and surfaces of environments afford different functions to a child, depending on his or her action capabilities [23,24]. Therefore, children experience the environment by it given meaningfulness in relation to themselves and the environment functionality [41,43]. In this way, children’s learning can be maximized in environments that provide them with rich contexts and a myriad of interesting possibilities [7]. It seems access to gardens or natural environments could be important for preschool children, as it can increase their opportunities for participation in deep and complex play, which is thought to be necessary for typical development [44].

As we saw in this study, younger children may obtain social advantages from green spaces as they provide an opportunity to socialize through group activities, exploration, and play, which are essential for ideal development [16].

### 4.1. MotorProficiency at Nature Schools vs. Kindergartens

While the concept of affordances has been applied to enrich the environment, in terms of children’s development [4,24], natural environments afford free space and different plays, which are substantial factors in children’s development [45].

Outdoor environments could be stimulating and motivating for children’s perceptions; for example, a lawn that provides a chance to run, roll, and engage in other motor skills [10,32,41,43]. Indeed, free playing is important for gross motor skill development, such as walking, running, and balancing. Furthermore, convenient spaces with natural outdoor materials also allow crucial practice of fine and gross motor skills [9,25,29,46]. Natural prospects with diverse physical features such as sloping grassy grounds, forest vegetation, stones in different colors and shapes, sand, and soil afford different play chances and increase the opportunity of detection, proficiency, and social interactions [8]. Fiortoft [47] stated that children’s activities in nature and the outdoors have a positive and significant effect on the components of children’s gross motor skills, including balance, coordination, jumping, agility, and strength. He also showed the great importance of a child’s play environment in his/her development, because the child’s activities are affected by the environment. In addition, Maynard and Waters [46] showed that an open and natural environment can increase children’s activities and thus increase their physical development. It seems playing in nature causes children to be more physically active due to environmental conditions, as well as the tools and natural materials in nature, and to be drawn to various movement challenges such as climbing, jumping, speed, and coordination. When children play in an organized environment, they do not have the opportunity to be active enough or to challenge their limbs. As a result, natural open environments develop children’s fine and gross motor skills. Ethier [10] stated that natural materials exist in nature, with special and unique properties (e.g., small, large, loose, soft, and hard). These substances stimulate children to experience certain sensations that are not present in other substances. With these properties, these materials help children develop fine motor skills [10,29].

### 4.2. Social Maturity at Nature Schools vs. Kindergartens

Vygotsky’s [48] sociocultural paradigm cited that a person’s knowledge is constructed by involvement and linking with various social environments. That is, social developmental key factors such as individual skills, creativity, and autonomy.

Although social development requires interaction with others, individual development (individual skills, creativity, autonomy, etc.) is still the basis of social development. For as much as the ability to partake could be achieved through the experience, so, participation of children in outdoor environments has beneficial effects. Additionally, from the viewpoint of pedagogy, children should participate actively in the world around them, including the outdoor environment, deciding and choosing what they want to do or learn [30,49]. It has been shown that the development and learning of children are associated with greater social play [25,50]. Repeated experience in nature allows children to have more self-confidence and to combine physical challenges in play. Interestingly, there is a deep relationship between the physical and social factors of children’s experiences. Social experiences such as a child’s individual responses to others or objects, their interactions with peers and teachers, their personal challenges, their responses to particularly difficult affordances such as rushing water or a steep rock, and their freedom of exploration are largely a function of the physical environment [38].

Moreover, there is a relationship between the social demands of childhood and behavioral, emotional, and cognitive integration [3]. If the social play environment is fun, peaceful, and dynamic (such as nature) and children feel safe, their interest in play and physical activity will increase [9,32]. Natural environments provide diverse, imaginative, and creative plays that stimulate children’s social interaction, communication, sharing, collaboration, independence, and coexistence. In the natural environment, children learn social skills such as how to behave and communicate with peers, self-confidence, and work ethic. In addition, interacting with natural elements helps children to discuss what they see and feel with those around them [47]. Bjorgen [31] stated that providing an environment that has social interactions is important to show the mutual needs of human beings. Social relationships, inviting children to play with one another, support reactions to a variety of behaviors, social roles, and social challenges posed by activities in the natural environment, in addition to improving a child’s relationships and increasing the duration of physical activity [29]. Additionally, simultaneously with the growth of a child, focus on the self gives way to an extended awareness of peers through social cooperation. Thus, providing more challenges and support, as well as providing each child with plenty of chances to encounter varying degrees of risk during play in sociocultural contexts in which there are decreasing chances for unlimited play, are very important [47]. In this regard, if facilitators encourage freedom of movement and afford opportunities for children to cooperate with one another in a challenging environment, children’s confidence, development, exploration, and social engagement will increase [39].

### 4.3. Limitations and Suggestions

One of the strengths of the present study was the quasi-experimental research method, but there are several limitations. Although the motor proficiency subscales were measured by an objective method, social development was imperatively collected by parent’s reports, which may have resulted in social desirability bias.

Moreover, in the present study, we could not survey the pleasure or satisfaction of children from activities in different types of spaces. In addition, our intervention led to positive results, but the development of interventions to improve the physical environments experienced by children across the globe is needed.

Although from an ecological viewpoint, the characteristics of the person, environment, and task are important to study, we could not control all of the variables in this study. The different kindergarten teachers, interactions with the children, the environment, the playing context, etc. could have affected the results. Thus, it is unclear whether the effects were due to children being outside, the physical space, their engagement in the various activities, or the kindergarten teachers’ behavior. Although children did the activities that could be done in each section on a completely voluntary basis, we could not record the time spent in each activity in the nature school.

## 5. Conclusions

The findings of the analyses highlight the significance of diverse and challenging environmental affordances for motor proficiency and social maturity. A developmentally rich context such as that of nature schools can provide assured, confident, and enriching opportunities for the development of almost all motor proficiency and social maturation of preschool children. However, the same effect of kindergartens on the mentioned factors was not observed. Unfortunately, the environments of many structured preschools do not have adequate opportunities and are designed from the viewpoint of adults, so they do not fit with children’s needs and desires [25]. Thus, pedagogical ideals suggest that many outdoor environments should be modified to respond satisfactorily to the needs of the children and to enrich their daily outdoor activity [31,43,51]. Therefore, multiple objects, natural and pristine spaces, animals, and interaction with peers should be provided. This is important because the person, the environment, and the task are tightly integrated during childhood [52]. Overall, motor proficiency is stimulated and achieved through appropriate challenges, and following improved motor skills, multiple and variable social and cognitive experiences become apparent. Thus, this issue should be explored in future research.

## Figures and Tables

**Figure 1 children-08-00157-f001:**
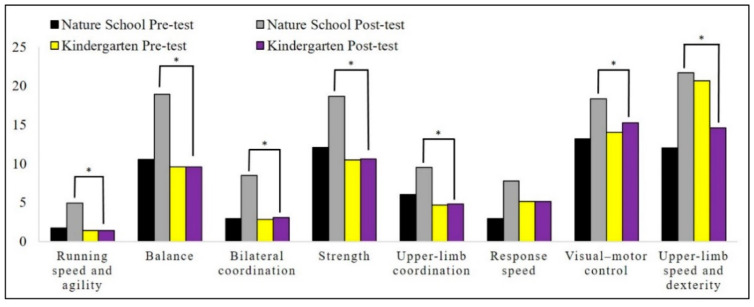
Descriptive statistics of the motor proficiency subscales. * *p* ≤ 0.05.

**Figure 2 children-08-00157-f002:**
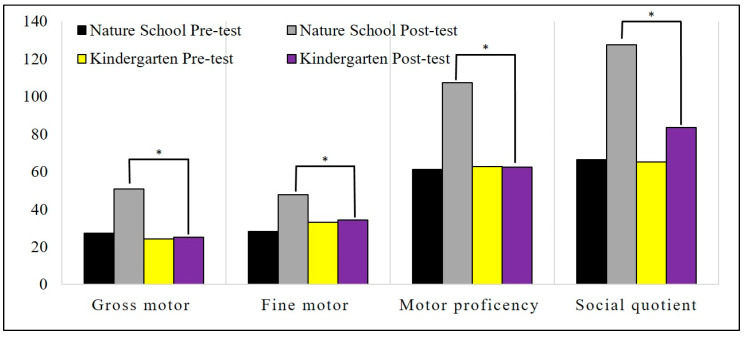
Descriptive statistics of the gross and fine motor proficiency and social quotient. * *p* ≤ 0.05.

**Figure 3 children-08-00157-f003:**
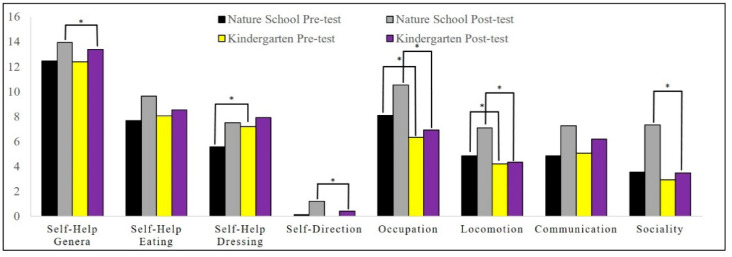
Descriptive statistics of the social development subscales. * *p* ≤ 0.05.

**Table 1 children-08-00157-t001:** Differences of the motor proficiencies between groups pre- and posttest.

Test Stage	Motor Proficiency	Mean Difference (Nature School–Kindergarten)	Std. Error	*p*
Pretest	Running speed and agility	0.33	0.62	0.599
	Balance	0.93	1.62	0.569
	Bilateral coordination	0.13	0.58	0.821
	Strength	1.60	1.54	0.310
	Gross motor skills	2.93	2.42	0.237
	Upper-limb coordination	1.33	1.27	0.303
	Response speed	−2.20	1.54	0.164
	Visual–motor control	−0.80	1.66	0.634
	Upper-limb speed and dexterity	−8.66	6.64	0.203
	Fine motor skills	−5.00	2.85	0.090
	Motor skills	−1.40	4.43	0.754
Posttest	Running speed and agility	3.53	1.13	0.004 *
	Balance	9.33	1.55	0.000 *
	Bilateral coordination	5.40	0.65	0.000 *
	Strength	8.06	1.52	0.000 *
	Gross motor skills	25.80	3.35	0.000 *
	Upper-limb coordination	4.73	1.25	0.001 *
	Response speed	2.66	1.56	0.099
	Visual–motor control	3.06	1.45	0.044 *
	Upper-limb speed and dexterity	7.13	1.44	0.000 *
	Fine motor skills	13.53	3.495	0.001 *
	Motor skills	45.06	8.204	0.000 *

* *p* ≤ 0.05.

**Table 2 children-08-00157-t002:** Differences of the motor proficiencies within groups pre- to posttest.

Group	Motor Proficiency	Mean Difference (Pre Test–Posttest)	Std. Error	*p*
Nature school	Running speed and agility	−3.20	0.750	0.0001 *
	Balance	−8.40	0.941	0.0001 *
	Bilateral coordination	−5.53	0.361	0.0001 *
	Strength	−6.60	0.816	0.0001 *
	Gross motor skills	−23.80	2.285	0.0001 *
	Upper-limb coordination	−3.53	0.520	0.0001 *
	Response speed	−4.86	0.568	0.0001 *
	Visual–motor control	−5.13	0.518	0.0001 *
	Upper-limb speed and dexterity	−9.73	4.820	0.053
	Fine motor skills	−19.73	1.339	0.0001 *
	Motor skills	−46.06	3.841	0.0001 *
Kindergarten	Running speed and agility	0.0001	0.750	0.999
	Balance	0.0001	0.941	0.999
	Bilateral coordination	−0.26	0.361	0.466
	Strength	−0.13	0.816	0.871
	Gross motor skills	−0.93	2.285	0.686
	Upper-limb coordination	−0.13	0.520	0.799
	Response speed	0.0001	0.568	0.999
	Visual–motor control	−1.26	0.518	0.021 *
	Upper-limb speed and dexterity	6.06	4.820	0.219
	Fine motor skills	−1.20	1.339	0.378
	Motor skills	0.40	3.841	0.918

* *p* ≤ 0.05.

**Table 3 children-08-00157-t003:** Differences of social development between groups pre- and posttest.

Test Stage	Social Development	Mean Difference (Nature School—Kindergarten)	Std. Error	*p*
Pretest	Self-help genera	0.067	0.52	0.898
	Self-help eating	−0.400	0.72	0.581
	Self-help dressing	−1.633	0.78	0.043 *
	Self-direction	0.133	0.10	0.153
	Occupation	1.767	0.39	0.000 *
	Locomotion	0.667	0.29	0.031 *
	Communication	−0.200	0.52	0.703
	Sociality	0.633	0.60	0.303
	Social quotient	1.433	6.80	0.835
Posttest	Self-help genera	0.533	0.24	0.038 *
	Self-help eating	1.100	0.79	0.174
	Self-help dressing	−0.433	0.83	0.604
	Self-direction	0.767	0.32	0.024 *
	Occupation	3.600	0.50	0.000 *
	Locomotion	2.767	0.31	0.000 *
	Communication	1.067	0.52	0.051
	Sociality	3.867	0.55	0.000 *
	Social quotient	4.014	10.84	0.000 *

* *p* ≤ 0.05.

**Table 4 children-08-00157-t004:** Differences of the social development within groups pre- to posttest.

Group	Social Development	Mean Difference (Pretest—Posttest)	Std. Error	*p*
Nature school	Self-help genera	−1.467	0.332	0.0001 *
	Self-help eating	−1.967	0.291	0.0001 *
	Self-help dressing	−1.933	0.403	0.0001 *
	Self-direction	−1.067	0.220	0.0001 *
	Occupation	−2.433	0.320	0.0001 *
	Locomotion	−2.233	0.177	0.0001 *
	Communication	−2.400	0.273	0.0001 *
	Sociality	−3.767	0.352	0.0001 *
	Social quotient	−61.178	4.987	0.0001 *
Kindergarten	Self-help genera	−1.000	0.332	0.005 *
	Self-help eating	−0.467	0.291	0.120
	Self-help dressing	−0.733	0.403	0.080
	Self-direction	−0.433	0.220	0.059
	Occupation	−0.600	0.320	0.071
	Locomotion	−0.133	0.177	0.458
	Communication	−1.133	0.273	0.0001 *
	Sociality	−0.533	0.352	0.141
	Social quotient	−18.597	4.987	0.001 *

* *p* ≤ 0.05.

## Data Availability

Data are contained within the article.

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
