# Peer review of "The Effect of Environmental Contexts on Motor Proficiency and Social Maturity of Children: An Ecological Perspective"

_children, 2021, doi:10.3390/children8020157_

Round 1

Reviewer 1 Report

Dear authors

It is a very interesting study under the assumption that natural outdoor physical activities would benefit several locomotor variables and cognitive function in children when compared to sedentary indoor activities. Overall the manuscript is poorly written which makes it difficult to fully understand the content and outcomes in which it was intended.  I have outlined some of my concerns in detail below.

Introduction

This section is too lengthy and also fails to provide a sound scientific rationale as to why this project was conducted. There are numerous misspelling and grammatical errors throughout the manuscript which must be corrected. Below, I have outlined some examples.  

Line 63- “broaddiversity” , “propertieswarrants”

Line 83- “probeenvironments”

Line 85- “approachconstructed”

Materials

How many participants in each group? How were they randomized?  no demographic info is presented.

Line 97- add country here.

Line 111- “getting dirty and make noise.” This does not sound much scientific.

Discussion

This section is also too wordy and lacks cohesion and reading flow in relation to the outcomes of the project.

Conclusion

Line 349-351-  “Unfortunately, the environments of many preschools (especially in developing countries such as Iran) do not have adequate opportunities and are designed from the adults’ viewpoint, sothey aren’t fit with children’s needs [25].”

This a strong statement with no proper reference. How is this relevant to your project?  I do not think natural playing requires significant infrastructure funding.  

Author Response

Dear Reviewer, 

See the attached answers.

Thank you!

Reviewer 2 Report

Interesting article, despite the theme being on the current educational agenda. Design consistent with appropriate methodological procedures (despite the difficulty of ensuring full control of variables). In the presentation of results, it is suggested that tables are accompanied with descriptions that best highlight a reading and help to "cut" the sequence of tables (too close together). Results and discussion (although there was some content here that looked more like the introduction part?).

Need to review jackdaws and other issues (spaces, etc.)

Author Response

(The authors gave the same response as above.)

Reviewer 3 Report

Overview:

The authors present a pre/post-test semi-experimental study examining the effects of “nature school,” or play outdoors, versus time spent in a typical kindergarten setting on 5- to 6-year-old motor skills on standard assessments (i.e., Bruininks-Oseretsky test of motor proficiency and Vineland scale). There results show children who were in the “natural outdoor play” group scored higher on several subscales for motor and social skills as compared to children in attending regular kindergarten. The authors argue that, from an ecological perspective, a natural environment stimulates play and supports optimal development for children. It is laudable that the authors sought to collect semi-experimental data testing effects of natural outdoor play versus (presumably) indoor play as physical activity and activities outdoors have been shown to affect children’s cognitive development. Studies that focus on children’s motor development beyond infancy are infrequent, particularly outside of the U.S. So, I was excited to review this work. However, several concerns about the study and manuscript need to be resolved.

  1. Introduction and framing of this paper needs work. Currently, the introduction is unreadable. It is vague, includes several theories that loosely relate to the data, and does not motivate the research questions or main variables of interest.

  1. The introduction is missing critical literature: empirical studies examining the importance of outdoor play and studies that have relied on these standard assessments in settings and cultures outside of North America.

  1. Critical aspects of the setting are missing. First, there is no description about the context in which children play. What do we know about kindergarten in Iran? It’s unclear where does natural outdoor play is occurring and how. Is it also part of the kindergarten curriculum ?

  1. The experiment has a fatal flaw, it’s confounded. It’s unclear whether the effects are due to children being outside, the physical space, or engagement in the various activities. For example, are children moving more when they are outdoors or when they are indoors? The only aspect of the “kindergarten” environment mentions activities such as painting and crafts, which involve children being stationary. But, just because they are indoors, does not mean they don’t engage in movement. So, it’s difficult to disentangle whether it’s the qualitative differences in the space or differences in the kind of behaviors children produce.

  1. No specifics are given about the two scales used in this study. These were developed and used with English-speaking children growing up in North America. The introduction/current study needs a motivation for using these metrics. The method needs specifics about how these scales were administered and whether any adjustments had to be made.

  1. The tables are unreadable. Recreate the tables rather than copy/paste from SPSS. The figures should be bar graphs rather than line graphs to show differences. Use asterisks to illustrate significance.

  1. The discussion has interesting points but none supported by data from the current study. For example, running, climbing the authors claim as important behaviors common during outdoor play are not described in this paper. Do children in this study do these behaviors?

  1. The Vineland scale is poorly motivated. How do the skills on the Vineland relate to children’s behaviors in the natural outdoor setting versus kindergarten setting?

I think that the author conducted an interesting study and the manuscript should be accepted, but I do feel that it could be strengthened if the researchers added important elements specified above.

Author Response

(The authors gave the same response as above.)

Round 2

Reviewer 1 Report

I would like to thank the authors for addressing my comments; the manuscript is much improved.

Author Response

Thank you very much!

Reviewer 3 Report

I gratefully recognize the authors’ effort to address the reviewers’ concerns. There are still several issues that concern me. Again, the attempt to conduct a pre/post experiment testing the effects of outdoor play is a strength of this study. However, the scarcity in the behavioral descriptions and the use of standard assessments to capture development are weaknesses. In this study, the authors are not testing outdoor vs indoor factor as a potential contributor to motor skills. Rather, the authors are comparing the different kinds of activities offered to children (e.g., painting vs running around). The authors do detect differences at the post-test session, but it is unclear to what we can attribute these differences. Importantly, we the authors offer no detail as to what activities children chose to engage in and for how long. Would it be possible for authors to consider coding the various behaviors children demonstrate when engaged in the indoor activities and when engaged in outdoor activities? Second major issues, the analyses are still unclear. Have the authors conducted one omnibus test comparing activity x group x session (pre vs post)? In the results section, it’s not clear whether time effects are being controlled within the same test that is comparing the effect of activity. Please clarify. Throughout the paper, language needs to be pared down. This study cannot address whether outdoor activity has an effect on motor development. The authors present some obscure differences between groups but the measures are not clear.

Author Response

Dear Reviewer, 

I attach our answers.

Thank you!
